# Acinetobacter Baumannii Phages: Past, Present and Future

**DOI:** 10.3390/v15030673

**Published:** 2023-03-03

**Authors:** Qihang Tu, Mingfang Pu, Yahao Li, Yuer Wang, Maochen Li, Lihua Song, Mengzhe Li, Xiaoping An, Huahao Fan, Yigang Tong

**Affiliations:** 1College of Life Science and Technology, Beijing University of Chemical Technology, Beijing 100029, China; 2Beijing Advanced Innovation Center for Soft Matter Science and Engineering (BAIC-SM), Beijing University of Chemical Technology, Beijing 100029, China

**Keywords:** *Acinetobacter baumannii*, bacteriophage, multi-drug resistance, phage therapy

## Abstract

*Acinetobacter baumannii* (*A. baumannii*) is one of the most common clinical pathogens and a typical multi-drug resistant (MDR) bacterium. With the increase of drug-resistant *A. baumannii* infections, it is urgent to find some new treatment strategies, such as phage therapy. In this paper, we described the different drug resistances of *A. baumannii* and some basic properties of *A. baumannii* phages, analyzed the interaction between phages and their hosts, and focused on *A. baumannii* phage therapies. Finally, we discussed the chance and challenge of phage therapy. This paper aims to provide a more comprehensive understanding of *A. baumannii* phages and theoretical support for the clinical application of *A. baumannii* phages.

## 1. Introduction

*Acinetobacter baumannii* (*A. baumannii*) is an essential Gram-negative pathogenic bacterium, widespread in nature [1]. *A. baumannii* can adhere to surfaces easily, as it has pods and pili [2]. Furthermore, since *A. baumannii* has strong invasive virulence factors, such as outer membrane proteins, lipopolysaccharides and phospholipases, the treatment of *A. baumannii* infection has been regarded as a great threat to clinical practice [3]. Antibiotics such as carbapenems, β-lactam antibiotics and polymyxins, are commonly used clinically to treat *A. baumannii* infections [4,5,6]. However, the treatment of multi-drug resistant (MDR) *A. baumannii* is further aggravated by the abuse of antibiotics and the evolution of bacteria. Bacteriophages are bacterial viruses that specifically recognize, infect, and replicate within the host bacteria [7,8]. Phages have been considered as therapeutic agents since the early 1920s as a result of their unique antibacterial ability. In addition, phages have the advantages of strong antibacterial ability, high quantity (10^30^–10^32^ in the earth), and low toxic side effects to humans, and are considered as one of the most promising drugs to replace traditional antibiotics [9,10].

## 2. Antibiotic Resistance in *Acinetobacter baumannii*

Drug-resistant bacteria continue to emerge, posing a huge challenge to human health and safety. Some studies have shown that antibiotic resistance genes (ARGs) of bacteria, such as trimethoprim resistance genes (*dfrA*), existed long before the clinical application of antibiotics [11]. *A. baumannii* is one of the most common opportunistic pathogens in nosocomial infections. Bacteria can develop antimicrobial resistance (AMR) to a variety of antibiotics, such as β-lactams [12], quinolones [13] and polymyxin [14], through intrinsic resistance mechanisms, such as increased efflux pumps [15], decreased outer membrane proteins (OMPs) [16] and acquired resistance mechanisms [14]. In addition, phages can also mediate antibiotic resistance in bacteria through transduction.

### 2.1. β-Lactam Class

β-lactam antibiotics (BLAs) are widely used in the clinical treatment of *A. baumannii* infection because of their ability to act on penicillin-binding proteins (PBPs) and inhibit the synthesis of cell walls [17]. However, the hydrolysis of BLAs by β-lactamase [18] and the structural changes of PBPs seriously affect the clinical efficacy of BLAs [19]. Carbapenem antibiotics include imipenem (IPM) and meropenem, of which IPM is the first highly effective broad-spectrum carbapenem [20]. Carbapenems are atypical BLAs and are considered to be one of the first choices for the treatment of *A. baumannii* infection [21]. However, since the first carbapenem-resistant OXA-type β-lactamase (*bla*_OXA-23-like_ enzyme) was discovered in *A. baumannii* strains, many β-lactamases that can hydrolyze carbapenems have continuously been discovered, such as *bla*_OXA-51-like_ enzyme and *bla*_OXA-58-like_ enzyme [22]. In addition, the reduction of OMPs can also lead to an increase in the resistance of *A. baumannii* to carbapenems [23]. With the increasing resistance rate of carbapenem-resistant *A. baumannii* (CRAB), how to treat CRAB has become a difficult problem worldwide. Previous studies have demonstrated that extensively drug-resistant (XDR) or pandrug-resistant (PDR) CRABs can lead to high morbidity and mortality, and the carbapenem antibiotic resistance rate has reached 90% in some regions [24]. The World Health Organization has identified CRAB as a prime pathogen for urgent drug development.

The *bla*_OXA_ gene is widely spread around the world. Kusradze Ia et al. conducted molecular tests on 12 *A. baumannii* isolates obtained from different countries, and the results showed that six strains containing the *ISAba1* and *bla*_OXA-51_ genes were resistant to the carbapenem antibiotic IPM. Moreover, it was found that the *bla*_OXA-23_ gene from the Iraqi isolates was located on the plasmid, while the *bla*_OXA-24_ gene from the Georgian isolates was located on the chromosome [12]. Horizontal transfer is an important way for *A. baumannii* to rapidly acquire antibiotic resistance genes, which is mainly mediated by transfer plasmids. Many plasmid-dif (pdif) sites were found in the plasmids of clinical isolates of *A. baumannii*, which are the targets of XerC and XerD recombinases and are believed to be conducive to the transfer of drug resistance genes [25]. Plasmids < 20 kb account for 56% of Acinetobacter plasmids, and plasmids > 20 kb carry ARGs that are usually encoded in other mobile genetic factors (MGEs), such as transposons, and it has been demonstrated that some plasmids > 20 kb contain genes related to phages [26]. This means that phage-mediated transduction is also one of the ways to obtain ARGs. In addition, ARGs such as carbapenemases also play a role in studying the evolution of the *A. baumannii* clade. For example, Hamidian M et al. used carbapenem resistance and aminoglycoside resistance genes to study the evolution of the clade of global clone 1 (GC1) lineage 1 [27].

### 2.2. Quinolone Class

Fluoroquinolone antibiotics inhibit DNA replication by targeting DNA gyrase and topoisomerase IV, thereby impeding bacterial growth. Through mutation analysis of fluoroquinolone-resistant *A. baumannii*, Geisinger E et al. proved that the mutation of *gyrA* and *parC* encoding DNA gyrase and topoisomerase IV, respectively, resulted in a change in the phenotype of the two enzymes, thus reducing the affinity of fluoroquinolone antibiotics for enzymes [13]. In addition, they found that the SOS response may enhance the fluoroquinolone resistance of *A. baumannii* by increasing the horizontal spread of ARGs or promoting the expression of genes, such as DNA repair and mutation.

### 2.3. Polymyxin Class

Polymyxin B (PMB) is considered to be the last line of defense against drug-resistant *A. baumannii* [28]. It changes the bacterial outer membrane (OM) charge by interacting with lipid A, causing an increase in the permeability of the OM and a disruption of the OM structure, thereby achieving the purpose of sterilization [29]. However, modification of lipid A by positively charged phosphoethanolamine (PetN) transferase and 4-amino-4-deoxy-l-arabinose (L-Ara4N) transferase can prevent the positively charged cationic region of PMB from binding to negatively charged lipopolysaccharide, resulting in resistance to PMB. Kim M et al. screened 40 clinical isolates of *A. baumannii* and found four anti-PMB strains with point mutations in the pmrB gene, leading to a high expression of the pmrC gene encoding PetN transferase [14]. In addition, they found that NCCP 16,007, which is more resistant than other PMB-resistant strains, obtains more ARGs from other pathogens through horizontal gene transfer. Moreover, *A. baumannii* can cause lipopolysaccharide loss and outer membrane remodeling by mutating the gene encoding lipid A and result in point mutations in the pmrA and pmrB genes of the two-component system pmrAB, respectively, and thus develop resistance to polymyxin [30]. Zhao et al. demonstrated that the level of resistance evolution of *A. baumannii* to PMB is related to the concentration of PMB, and that higher concentrations of PMB are more favorable for the evolution of bacterial resistance [31].

### 2.4. Phage-Mediated Antibiotic Resistance

Acquired resistance of *A. baumannii* is usually achieved by bacterial conjugations. In addition, phage-mediated transduction is an important way to obtain ARGs. By analyzing the genomes of 177 prophage strains of *A. baumannii*, Loh B et al. found that some prophages carried resistance genes, such as *bla*_OXA-23_ and *bla*_NDM-1_ [32]. It has been demonstrated that phages can introduce ARGs through transduction. The plasmid pABTJ2 of MRAB MDR-TJ has been detected to contain many phage-like elements [33]. Some resistance genes have been found to be integrated into chromosomes. In this regard, Wachino JI et al. proved that drug resistance genes in *A. baumannii* could be transmitted by prophages without direct interaction between cells [34]. To explore the intraspecies transmission of the carbapenemase gene *bla*_NDM-1_, Krahn T et al. paired an *A. baumannii* donor strain R2090 with recipient strain CIP 70.10, to obtain a carbapenem-resistant derivative [35]. Moreover, it was confirmed that R2091 received the transposon Tn125 containing the *bla*_NDM-1_ gene, and it was speculated that the activation of a prophage in the genome of strain R2090 could promote the transduction of the carbapenem gene *bla*_NDM-1_.

## 3. Acquisition and Characterization of the *A. baumannii* Phage

The isolation, identification and characterization of *A. baumannii* phage biology is the first step in phage therapy, and related research is summarized below.

### 3.1. Source

Bacteria are the most numerous biological entities on Earth, which outnumber their hosts. Phages are widely distributed in a variety of ecosystems. The number of phages in the aquatic environment is approximately 10^4^–10^8^ PFU/mL. The highest abundance of phages is found in coastal waters, at approximately 10^6^~10^7^ PFU/mL [36]. Phages are present in both animals and humans, with over 10^8^ phages per gram of feces [37]. Sources of *A. baumannii* phages include hospital effluent, lesions and the sputum of patients and birds from free-range farms. Hospital effluent is the main source.

### 3.2. Structure and Genomics

Normally, the head of the phage is prismatic, and the single genetic material (DNA/RNA) is contained, enveloped by protein. Attached below the head is the neck or collar region of the elongated sheath (sometimes called the tail). The DNA/RNA is injected into the host cell through its internal hollow structure and is surrounded by protective sheath proteins [38]. The base of the sheath is a baseplate to which the tail fibers are attached, which is a key structure for the attachment and infection of host cells, and its function is primarily to recognize the surface receptor of the host cell and complete the infection process by binding to the host cell receptor. The tail fiber protein is typically composed of multiple subunits and has high variability, which allows the formation of diverse tail fiber structures by different combinations of subunits [39], enabling the infection of different host cells, thereby tail fiber proteins can be used in the detection of *A. baumannii* [40,41,42]. The special cases include filamentous phage [43] and barely non-tailed phage [44]. At present, there are no definite reports that the host of filamentous phage is *A. baumannii*.

The viral nature of phages was controversial until the early 1940s, when they were observed by electron microscopy, confirming their particulate nature and enabling them to be classified based on morphology [45]. Classic electron microscopy images are formed by atoms of heavy metals such as uranium, which evaporate in a vacuum and allow for the sample to be struck at an angle. The introduction of negative stains (heavy metal salts that dry in thin layers, do not form crystals and can embed small particles, such as phages) in electron microscopy has resulted in more detailed images than earlier methods and revealed the complexity and diversity of phage morphology [46]. Assigning phages to different taxonomies is a fundamental step in phage research. As more and more new phages are discovered, ICTV’s classification criteria are constantly changing. The most recent standard is the August 2022 phage classification system, which removes several major families in the order Siphoviridae, Podoviridae and Myoviridae. However, the classical description of its morphology as belonging, such as “podovirus”, “myovirus”, or “siphovirus”, remains. The order “Caudovirales” was also deleted and replaced by the class Caudoviricetes, and a binomial system of nomenclature for species was established [47].

Genome sequencing revealed the abundance of the prophage. Comparative genomics showed the co-evolutionary relationship between phages and their host bacteria as an essential tool to reveal phage diversity [48] and provide the strategy of therapy [49]. The framework of the *A. baumannii* phage genome is mainly composed of genes from other phages, with a relatively small proportion of genes from the host *A. baumannii* [50]. Morphogenesis of the tail occurs frequently, the mutant Ab105-2phiΔCI404ad, genomic rearrangement increases the host range of the phage nearly 3-fold [51]. There is no genome similarity of vB_AbaS_TCUP2199 to other known phages [52]. It has been documented that most of the genes of bacteriophages are unknown [53]. Their genome size range is wide, and the structure of their genome is linear or circular [54]. The most common type of genome is dsDNA among known phages. The genome size of phage KARL-1 was determined to be 166,560 bp. A total of 253 ORFs were identified, involved in the replication, maturation and release of phage progeny. Whereas some encoded structural proteins, most ORFs encoded hypothetical phage-like proteins, and the rest are hypothetical proteins [55]. The genome of *A. baumannii* phage Abp9 contains 80 ORFs, but lacks any known virulence genes or lysogen-forming genes [56]. The two phage strains, WCHABP1 and WCHABP12S, contain seven structural proteins. In addition, both encode a gene for a protein-containing lysozyme that is also possessed by other phages of the genus Ap22virus [57].

### 3.3. Life Cycle and Biological Properties of Phages

Phage growth is determined by many parameters, namely, the phage bacterial adsorption rate constant, burst size, latent period, bacterial growth rate, phage and bacterial elimination rates, and the effect of controlled release of the phage [58,59].

Phages can be divided into temperate phages and lytic phages, which have different life cycles (Figure 1). Infection begins with the adsorption of phage, which interacts with specific receptors on the surface of host cells, and then the phage injects its DNA into the cytoplasm of the host, next, transcription and replication occur. Subsequently, once the synthesis and assembly of viral proteins are complete, the phage DNA is packaged into the capsid. Phage-encoded depolymerase hydrolyses the peptidoglycan layer of bacterial cells, leading to cell lysis and the release of mature virions. The phage progenies are released into the environment and they can then infect the next phage-sensitive receptor [60]. Temperate phages can enter either lytic or lysogenic cycles [61], for example, λ phage, after the injection of DNA into the cytoplasm. The temperate phage could choose to initiate a lytic cycle, which would be consistent with the virulent phage. It integrates its DNA into the host bacterial chromosome. At the same time, the viral genome is named a prophage. This phage expresses a specific phage repressor that represses transcription factors, including lytic cycle genes, and thus the phage enters a dormant state [62]. The integrated phage dsDNA replicates with the bacterial chromosome during cell division, and is thus passed down through generations in bacteria. When the phage is excised from the host chromosome, it exits the lysogenic cycle [63].

The biological properties of different phages differ. vB-AbaM-IME-AB2 was able to adsorb host cells within 9 min with >99% adsorption, a 20 min latency period and a small outbreak size (62 PFU/cell) [64]. Two phages (WCHABP1 and WCHABP12) were obtained from hospital sewage, with different lysis ranges and burst sizes of 136 and 175 PFU/cell, respectively, with 99% adsorption within 10 min, and the MOIs of both were 0.1 [57]. Phage PD-6A3 had better activity at temperatures of 4–50 °C and from pH 5 to 10, with 90% adsorption within 5 min [65]. The clinical isolate PlyF307 is the first highly active therapeutic-specific *A. baumannii* phage against a Gram-negative bacterium, and rescued 50% of mice in a mouse animal model [66]. For Gram-positive bacteria, the rescued rates are usually higher [67,68].

## 4. Action of *A. baumannii* Phage on *A. baumannii*

### 4.1. Phage Recognition

The phage life cycle consists of five stages: adsorption, infestation, replication, assembly and release, the most important of which is the adsorption process [69] (Figure 2). Phage tail recognition sites are abundant and diverse, such as lipopolysaccharides, lipoproteins and capsules of *A. baumannii*, which are potential phage recognition sites. Phages expressing depolymerase degrade bacterial outer polysaccharides and promote phage recognition and initial adsorption to the host [70,71,72].

The capsule is an important recognition site for *A. baumannii* phage and consists of capsular polysaccharide that wraps around *A. baumannii* and provides protection at the periphery [73,74,75,76,77,78]. Fernando et al. found that 24 strains of *A. baumannii* developed resistance to *A. baumannii* phage through evolution, and 20 of the phage-resistant strains showed capsule defects. Genetic comparison of the two resistant strains (AB900 and A9844) with the wild type, that both phage-resistant strains had a base deletion at the K site, resulting in a code-shifting mutation. The results showed that the gtr29 gene (encoding glycosyl transferase) was affected in AB900, and that the gpi gene (glucose-6-phosphate isomerase) was affected in A9844, causing capsule deletion [79]. Popova et al. used tail spike depolymerase to degrade capsular polysaccharide layers surrounding *A. baumannii* host cells and observed that specific phages could not adsorb to the cells. The results obtained confirm that the capsular polysaccharides are the primary receptors for the phages and that tails pike depolymerase plays a crucial role in the initial step of phage–bacterial cell interaction [80]. There is no doubt that phage infestation of *A. baumannii* inevitably disrupts the capsule, regardless of whether the phage receptors are capsules. In basic research data on the mechanism of phage-specific recognition and hydrolysis of capsules, studies on *A. baumannii* phages emphasize the type of capsule of *A. baumannii* that can be lysed [76,77,80,81,82,83].

With the invention and development of protein expression technologies and modern detection techniques, studies on phage and host recognition have been continuously reported. The tail fiber proteins gp52 and gp53 of *A. baumannii* phage AbTJ, labeled with fluorescein isothiocyanate, were combined with magnetic beads and the BL method for *A. baumannii* detection. The tail fiber proteins were observed to be bound to the surface of *A. baumannii* under fluorescence microscopy, but did not have a lytic function [84].

Arms races between bacteria and their phages promote each other to evolve for survival. Bacteria are forced to pay a price for phage threat evolution, especially when phage recognition sites are important to them. Phage-resistant strains are shown to suffer from adverse conditions, such as weakened metabolism, reduced virulence (capsule, virulence factors) and restored susceptibility to antibacterial substances (antibiotics) [79].

### 4.2. Phage Lyse Bacteria

Another critical stage of phage growth and reproduction is release. Phage synthesis of lysis-related functional proteins causes bacterial cells to rupture, thereby releasing the offspring. Phage therapy is a promising alternative route against the growing backdrop of severe global antibiotic resistance, but rapid bacterial resistance to phages limits the development of phage treatment. In the search for a breakthrough, the associated phage releases phase proteins that destroy bacteria and are considered to be a highly promising AMR agents. Therefore, the study of phage lysis bacterial pathways is imperative. This section focuses on the phage lysis pathway of *A. baumannii*.

In 1992, the classic holin–endolysin theory proposed that bacteriophages mainly rely on the synergistic cleavage of holin and endolysin proteins [60]. Holin protein is a kind of small molecular membrane protein that is synthesized in large quantities in the process of bacteriophage synthesis, to control the time of cleavage. Endolysin protein is a protease that hydrolyses peptidoglycan, which destroys the membrane structure of bacteria. Because of the comprehensive characterization of the lambda phage, the cleavage model is based on the lambda phage [85]. In the phage synthesis stage, holin protein is continuously synthesized and accumulates in the cytoplasm and intima in the form of a dimer. At the same time, endolysin protein is folded to form peptidoglycan-lytic enzymes and accumulates in the cytoplasmic matrix. When the holin and endolysin proteins accumulate to a critical value, holin “triggers” are triggered (in the lambda system, 50 min is triggered after the synthesis of holin and endolysin begins) [86,87]. The holin protein inserted into the intima randomly forms a nonspecific channel or hole that allows for the endolysin protein to pass through to cleave the peptide polysaccharide layer (Figure 2A). A sufficient amount of holin protein destroys the electrochemical balance of the membrane and causes local depolarization to form channels or pores. Studies by Gründling [88] have shown that the holin system can be triggered in advance by dinitrophenol (DNP) or sudden hypoxia, but premature cleavage is accompanied by a significant decrease in phage titer [89]. Due to the local action of the holin protein, the protuberance of bacterial cells at the action site of the holin protein was observed under a video microscope, followed by rupture that released substances in the cytoplasmic matrix [90]. A great deal of evidence shows that the holin proteins P2 Y and T4 T are different from the lambda system [91], the time of the holin “trigger” is determined by the holin protein itself [92,93,94,95,96].

With the deepening of molecular-level research, a new cleavage pathway, namely, the SAR endolysin and pinholin pathways, has been found [97]. The infiltration of SAR endolysin into the intima is not dependent on holin protein: it permeates the intima in the form of a membrane plug. SAR endolysin accumulates in the intima, but is not released, and the membrane plug form does not have the enzymatic activity to avoid premature opening and cleavage, leading to a low phage titer. When the molecular dynamic potential of the intima changes, such as in depolarization, SAR endolysin can be released from the intima to the peptidoglycan layer, and the enzyme active center can be constructed by folding, to catalyze the hydrolysis of the peptidoglycan layer [98,99]. The role of changing the dynamic potential of membrane molecules and control of the timing of cleavage could occur via a kind of holin protein that is not directly involved in the output and is only used as a timer [100]. To distinguish it from the holin protein with an output function in the classic holin–endolysin pathway, this kind of protein is called pinholin. Pinholin accumulates in the intima, and when the time for cleavage comes, it polymerizes into regular heptamer channels, destroys the membrane dynamic molecular potential, and promotes SAR endolysin to obtain activity and cleave peptidoglycan [101] (Figure 2B). In addition, studies have shown that pinholin cannot release classic endolysin, which further proves that holin–endolysin and SAR endolysin/pinholin are two different cleavage pathways [102]. Both pathways lead to bacterial lysis, but different rupture processes were observed under the video microscope, and the two can be clearly distinguished. As mentioned above, there are significant local protuberances in the holin–endolysin pathway, which is due to the massive release of local endolysin, due to the formation of channels or holes by holin. SAR endolysin is uniformly distributed on the intima. When pinholin triggers the cleavage cycle, SAR endolysin releases and cleaves peptidoglycan from all directions. Therefore, the gradual contraction and final cleavage of bacteria can be observed under a video microscope.

Before 2012, based on the study of the above two pathways, it was speculated that the outer membrane is not a barrier blocking the release of the progeny phage. However, Berry et al. described a class of cleavage proteins called spanins, indicating that for most Gram-negative bacteria, the peptidoglycan layer and outer membrane are barriers that bacteriophages need to break through [90]. In their study, a mutant without spanin function was constructed. The morphological protuberance of the bacteria was observed and spread around, finally forming a ball that did not release intracellular material. At present, the characterized spanin systems are divided into two categories: (i) i-spanin and o-spanin two-component systems and (ii) u-spanin single-component systems [103]. The action mechanism of spanins is still in the exploratory stage. Young R provides a reasonable mode of action [104]. He proposed that spanins lead to the final lysis of bacterial cells through membrane fusion. The most suggestive elements for a membrane fusion model, besides the possible fusogenic character of purified Rz1 reported previously, are the primary structure of the Rz periplasmic domain and the conformational dynamics attendant to spanin complex formation [105]. Based on the results of Bryl et al., the binding of the Rz1 protein to the Rz homodimer leads to a substantial increase of alpha helix as well as the formation of coiled-coil oligomers, which evokes the coiled-coil dynamics of the SNARE system that is integral to trans-Golgi vesicle fusion [106,107]. SNARE proteins undergo coiled-coil oligomerization to bring the vesicle membrane and cell membranes into contact [108]. In the hypothesis of Young, in the two-component system, i-spanin is connected to the intima, o-spanin is tied to the adventitia, and the peptide polysaccharide layer is clamped to prevent folding and contraction. When endolysin or SAR endolysin releases cleavage peptidoglycans, their conformations change, and folding contraction may bring the intima and adventitia closer and lead to membrane fusion. Membrane fusion could destroy the molecular potential of the inner and outer membranes, leading to the collapse of the inner and outer membranes. Based on the data by Bryl et al., it is speculated that the factor promoting membrane fusion may be o-spanin [105]. These hypotheses do not seem to be suitable for the one-component system of u-spanin, but membrane fusion is the preferred model for u-spanin at present (Figure 3).

Generally, the phage cleavage process of Gram-negative bacteria needs to break through three barriers: the intima, peptidoglycan layer and outer membrane. The cleavage of the intima and peptidoglycan layer is accomplished by two groups of matching proteins, namely, classical holin–endolysin and SAR endolysin/pinholin [104]. On this basis, the cleavage of the outer membrane at the spanin protein is an essential step [105]. At present, the characterized spanin system is divided into a u-spanin one-component system, and an i-spanin and o-spanin two-component system. In theory, there are four different combinations of bacteriophages in the cleavage of Gram-negative bacteria.

### 4.3. Phages Affect Bacterial Genomes

Bacteriophages have been widely recognized as natural and efficient carriers of certain bacterial toxins or resistance genes, including classic type I membrane-acting superantigens, type II porogenic lysin and type III exotoxins, such as diphtheria and botulinum toxin. In addition, there is a kind of effector protein encoded by bacteriophage in Gram-negative bacteria, which is a new type of bacterial virulence factor [109]. The life cycle of temperate phage is lysogeny. Lysogenic bacteriophages integrate their own genome into the host genome through integrase, replicate with the host genome and form offspring. Through this pathway, lysogen phages participate in the variation related to bacterial virulence or drug resistance [110,111], carrying genes encoding virulence (strong extracellular toxins), proteins related to promoting bacterial invasion, and various enzymes (superoxide dismutase [112], grape kinase, phospholipase, DNA enzyme, proteins that affect serum resistance and change antigenicity, superantigens, adhesion molecules, proteases, and mitogenic factors) [37,109,113,114]. In addition, the prophage genome can also bring quorum sensing and motility to bacteria [115]. The above phenomenon, which gives virulence to bacteria by bacteriophages, is called lysogen transformation [116,117]. Under the influence of inducers or adverse external factors (such as antibiotics), lysogen bacteriophages will enter the lytic cycle and produce offspring to find the host again. The induction and mobility of prophages promote the widespread of virulence genes, ARGs and mobile genetic elements and promote the evolution of bacteria [79,118,119].

Based on the genomic analysis of the current pandemic carbapenem-resistant *A. baumannii*, the Oxa23 gene related to drug resistance was detected and analyzed by Abouelfetouh et al. The results have shown that eight of the 13 strains carrying the Oxa23 gene were located on the genome of the phiOXA prophage [111]. Phages with the Oxa23 gene were successfully isolated by mitomycin C induction. Transposon or plasmid-mediated drug resistance is a common mechanism, but in recent years, it has been reported that prophages lead to drug resistance, which may be another main mechanism of the horizontal transfer of these genes [120,121,122,123,124]. Similarly, based on the analysis of the evolution of *A. baumannii* in the clinical ward over 10 years, it was found that lysophage provided *A. baumannii* with virus defense proteins and quorum sensing functions [115].

Phages interact with bacteria while using bacteria for reproduction and provide bacteria with genes that are conducive to bacterial survival (ARGs, virulence genes, quorum sensing, etc.). This is a very interesting phenomenon; bacteriophages seem to protect their “food” from being naturally eliminated. In addition, it has been reported that the release of ARGs by phage lysis can be introduced into nondrug-resistant *E. coli* and produce related drug resistance [125].

### 4.4. Phages Act on Biofilms

*A. baumannii* with biofilms are a collective community cluster formed by the accumulation of a large number of bacteria, surrounded by self-secreted fibrin isopolymers. A large number of studies on hospital clinical outbreaks, severe infections and antibiotic resistance of *A. baumannii* have identified biofilm formation ability as the main virulence factor of *A. baumannii*, while providing strong adaptability and resistance to antibiotics [126,127,128,129,130,131]. In clinical practice, *A. baumannii* forms biofilms on the surfaces of hospital equipment or on retention equipment in patients [132], especially in urinary tract infections [133,134]. Biofilms can effectively inhibit antibiotics elimination of bacteria, which is one of the important reasons for the high mortality of A. baumannii infection [135,136,137,138,139]. Biofilm-producing *A. baumannii* has become a serious challenge for clinical infection, and many chemical-, physical-, and biological-based methods have been developed to prevent or destroy biofilms [140,141,142,143]. However, bacteriophages that directly degrade bacterial biofilms are the most promising therapeutic method to solve this problem, so the study of phages acting on biofilms is crucial [144,145,146,147,148,149,150,151].

Phages act on biofilms, destroying the biofilm matrix, exposing receptors on the bacterial surface, and initiating the phage replication cycle. In addition, phage lysis of biofilms also increased the permeability to antibiotics, restoring the eradication effect of antibiotics on bacteria [152] (Figure 4). By comparing the antibiofilm activity of a single phage and a phage cocktail, it was found that the antibiofilm activity of the phage cocktail is significantly higher than that of a single phage [51]. Studies have shown that the combination of two different types of antibacterial agents, bacteriophage and antibiotics, is more effective than the use of a single component, and can eradicate biofilms [134,153]. Changes in the biofilm structure, caused by one or both of the antimicrobial agents, may account for the enhanced disinfection effect. For example, bacteriophages can make bacteria more sensitive to bacteriophages and certain antibiotics by disrupting their external structures and improving their internal metabolic state. Antibiotics themselves may also cause changes in the biofilm structure, thereby increasing the ability of phages to invade biofilms [152].

Phage disruption of biofilms relies on hydrolases such as phage endolysin and depolymerase, and Lu et al. characterized that endolysin PlyF307 significantly reduced plankton and biofilms of *A. baumannii* both in vitro and in vivo. PlyF307, the first highly active lysin protein against Gram-negative bacteria, rescues mice from lethal *A. baumannii* bacteremia. The endolysin belonging to other families can also lyse *A. baumannii* [154,155]. By comparing the engineered phage expressing depolymerase with the phage without depolymerase, it was found that the engineered phage significantly destroyed the biofilm of *E. coli* [66]. After Zhang et al. treated *A. baumannii* with bacteriophage AB3 and its LysAB3 for 24 h, the *A. baumannii* biofilm was significantly degraded [156].

As bacteriophages continue to be discovered and characterized to dissolve biofilm-related proteins, numerous studies have reported the use of bacteriophage endolysins or depolymerase for the efficient elimination of *A. baumannii* [157,158,159]. These antibacterial proteins do not have the advantage of self-replication, but are more difficult to tolerate than phages and less likely to elicit the body’s immune defense. Similar to bacteriophages, bacteriophage protein antimicrobials work better with antibiotics. In the clinical eradication strategy of bacteria, the use of phage-associated protein antibacterial strategies has great potential and application prospects.

### 4.5. Transcriptomic Analysis

Transcriptomics investigates gene transcription and its regulation at the overall level and detects gene expression at the RNA level. At present, most of the research on the action of *A. baumannii* phage on *A. baumannii* is limited for characterization and local protein mechanism research, and the global analysis of phage and host interactions is lacking. Transcriptomic alignment analysis can be used to better understand the effect of phages on bacteria and provide a reference for the development of phage therapy [160].

Yang et al. analyzed the differential expression of genes after the infection of host AB1 by *A. baumannii* phage φAbp1 using global transcriptome analysis. The alignment showed only 15.6% (600/3838) of the host genes were differentially expressed; that is, only a small part of the bacterial genome can be used to complete phage reproduction. Through gene co-expression networks, an intermediate protein, gp34, was found to be involved in RNA polymerase synthesis and negatively interacts with many host ribosomal protein genes, which suggests that gp34 may be a key gene that inhibits or shuts down the translation process of the host. This also means that the replication and translation process of the phage overtakes the host cell. In phage late proteins, downregulation of both phage assembly and biosynthesis resulted in a lack of the expression of phage structural proteins. In addition, upregulation of host proteolytic proteins was observed, suggesting that the host has a defense mechanism against phages. Based on GO and KEGGplus network analysis, despite the challenge of the host stress response, the regulation of phages remains precise and powerful. In bacteria, phage infection suppresses the expression of relevant virulence genes, but activates resistance genes. The underlying mechanism of this phenomenon is still unclear, but it presents new challenges for the clinical use of phage-antibiotic combination regimens [161].

The global analysis of transcriptomics provides a basic research direction for studying the interaction between phages and bacteria. At the same time, data analysis based on transcriptomics can judge the treatment potential of phages from a more holistic perspective, to analyze the changes in virulence and drug resistance after bacterial infection and select appropriate phages to formulate treatment plans. At present, transcriptomic analyses of *A. baumannii* phages and their hosts are very scarce. Although many reports have verified the antibacterial potential of phages in vitro or in vivo, there are no data to analyze the changes in phages and bacteria globally [162,163,164,165,166,167,168,169]. This brings a huge workload to the clinical formulation of phage therapy or phage-antibiotic combination therapy.

## 5. Bacteriophage Resistance Mechanism of *A. baumannii*

In the tens of millions of years of struggle between bacteria and phages, bacteria have evolved a variety of defense systems [170] to resist phage infection, and anti-phage mutant strains have been found constantly. Ambroa A et al. conducted whole-genome sequencing analysis on 18 clinical strains of *A. baumannii* and identified 118 to 171 genes related to phage resistance, including the abortion infection (Abi) system, CRISPR-Cas system, and restricted-modification (R-M) system [171]. Their research suggests that the number of genes associated with resistance to phages may be growing year by year (Figure 5).

Phage therapy, as an alternative therapy for multi-resistant *A. baumannii* (MRAB), extensive drug-resistant *A. baumannii* (XDRAB) and pandrug-resistant *A. baumannii* (PDRAB) infections, has gradually become the focus of attention. A more comprehensive understanding of the resistance strategy of *A. baumannii* against phage attacks is helpful to evaluate the antimicrobial ability of phages. In addition, according to the molecular mechanism of bacterial and phage resistance to each other, it is helpful to develop new phage therapies to solve the increasingly complex problem of antibiotic resistance and avoid the influence of the bacteriological anti-phage strategy on phage therapy.

### 5.1. Adsorption Inhibition

Inhibition of phage adsorption by phage receptor deletion or structural change is the first step taken by *A. baumannii* to defend against phage infection. Phages are recognized with host surface receptors by receptor-binding proteins (RBPs) typically located at the end of phage tail fiber [73]. The capsule of *A. baumannii* is the most common receptor for phage [79,166,172,173].

Phages ΦFG02 and ΦCO01 can infect *A. baumannii* strains AB900 and A9844, respectively. Gordillo Altamirano F et al. found that both strains, AB900 and A9844, had single nucleotide deletions at the K locus (capsule biosynthesis locus) [174] after coincubation with phages, which affected the gene *gtr29* encoding transferase and the gene *gpi* encoding glucose-6-phosphate isomerase, respectively. This resulted in the production of phage-resistant mutants, ΦFG02-RAB900 and ΦCO01-RA9844, with missing capsules. As a result, the adsorption of phages ΦFG02 and ΦCO01 to the strain was interrupted, which hindered the infection of the phage with *A. baumannii* [79]. Similarly, single-base deletions in the *gtr6* gene, which encodes glycosyltransferase in *A. baumannii*, AB5001, resulted in a change in the structure of the K3-type capsular polysaccharide (CPS), thereby inhibiting the ability of the phage to infect the bacteria [172]. The adsorption test proved that the capsular membrane was the main receptor of phage Phab24, while the outer membrane was the secondary receptor [166]. Wang X et al. obtained phage-resistant mutant strains with the changed structure of the capsular and extracellular membrane due to gene mutations, by coculturing phage Phab24 and *A. baumannii*. Importantly, both studies found that phage-resistant mutant strains were resistant to antibiotics due to the absence of a bacterial capsule. Among them, Gordillo Altamirano F et al. demonstrated that phage-resistant strains ΦFG02-RAB900 and ΦCo01-RA9844 were more sensitive to cephalosporins and β-lactams. They also found that strains resistant to Phab24 increased their sensitivity to colistin [166]. These findings support the combination of bacteriophages and antibiotics in the treatment of *A. baumannii* infections.

### 5.2. CRISPR-Cas in Acinetobacter baumannii

CRISPR-Cas consists of clustered regularly spaced short palindromic repeats (CRISPR) and CRISPR-associated (*cas*) genes, and is an acquired immune system [175]. It can be divided into the class 1 CRISPR-Cas systems (including three types, I, III and IV), where the effector complex consists of multiple Cas proteins, and the class 2 CRISPR-Cas systems (including three types, II, V and VI), where the effector complex consists of a single Cas protein [176]. The CRISPR array consists of the spacer from exogenous gene sequences and repeat sequences. CRISPR-Cas systems insert the protospacers of phage DNA into the CRISPR array. When the phages invade again, CRISPR-derived RNA (cr-RNA) induces the Cas proteins to cut the target genes, thus preventing the phage from infesting the bacteria [177].

There is an evolving and complex CRISPR-Cas system in the genome of *A. baumannii* [122]. At present, the I-F CRISPR-Cas system has been found in *A. baumannii*. Tyumentseva M et al. compared the strains carrying the I-F1 and I-F2 CRISPR-Cas systems, and found that the *A. baumannii* strain with the I-F2 system had more CRISPR arrays and stronger resistance to phage invasion than the *A. baumannii* strain with the I-F1 system [178]. In addition, they found that the CRISPR-Cas system appeared to be associated with virulence factors, with a higher proportion of virulence genes in strains that lacked the CRISPR-Cas systems. Karah N et al. investigated the CRISPR-Cas isoform I-Fb locus in 76 *A. baumannii* isolates from 14 countries and found that the locus was derived from a common ancestor. In addition, they demonstrated that CRISPR-based methods could be used for *A. baumannii*. These isolates can be divided into 40 CRISPR-based sequence types (CSTS), with the CST1 isolate originating from Iraq and three CST19 isolates originating from Thailand [179]. In addition, the CRISPR-Cas system belonging to the CST8 subtype was also detected in the *bla*_OXA-23_-containing *A. baumannii* ST409 isolated from Greece by Galani V et al. [180]. Mangas EL et al. performed genomic analysis of nearly 2500 *A. baumannii* and demonstrated that a group of strains with fewer plasmids (717 strains) had more CRISPR array and cas genes. This may be because the CRISPR-Cas system prevents plasmids from entering the bacteria. Moreover, they also found that this group of strains had more genes associated with the biomembrane formation [181].

Through the CRISPR-Cas system, bacteria can not only resist phages, but also obtain more ARGs. Some studies have shown that *A. baumannii* isolates with the CRISPR-Cas system usually carry more ARGs and have a stronger drug resistance [182]. At the same time, with the specific recognition property of the CRISPR-Cas system, it can be used as a tool to knock out, modify or silence drug-resistance genes in the bacterial genome, which is a new scheme for treating drug-resistant bacteria [183,184]. However, not all genes associated with CRISPR-Cas contribute to bacterial resistance. To explore the relationship between the csy1 gene in the type I-Fb CRISPR-Cas system and A. baumannii resistance, Guo T et al. knocked out the csy1 gene of A. baumannii AB43, and found that the AB43Δcsy1 mutant showed antibiotic resistance. Transcriptome analysis proved that the csy1 gene could lead to the antibiotic sensitivity of *A. baumannii* by regulating related genes, such as the Cas protein and efflux pumps [15]. Although the CRISPR-Cas system is ubiquitous in *A. baumannii*, not all *A.-baumannii*-resistant strains have the CRISPR-Cas system. For example, *A. baumannii* NCIMB8209, which contains *bla*_OXA-51-like_ genes on its chromosomes, has a low ability to form biofilms and membranes and a slow movement rate. Whole-genome sequencing analysis showed that the NCIMB8209 genome does not contain the CRISPR-Cas system [185].

### 5.3. Quorum Sensing

Quorum sensing (QS) is an intercellular communication system that can coordinate population density and regulate gene expression through QS signals, which helps *A. baumannii* respond quickly to environmental changes [186]. QS can promote the production of *A. baumannii* biofilms [187], which can protect bacteria from phage infection by shading phage receptors and other ways. *A. baumannii* possesses an *abaI*/*abaR* QS system, with n-acyl-homoserine lactone (AHL) as a signaling molecule, which consists of an AHL synthase (encoded by *abaI*) and a transcriptional regulator (encoded by *abaR*). The *abaI* and *abaR* genes are thought to have been obtained by *A. baumannii* from Thiobacillus neapolitanus by horizontal gene transfer [188]. The *abaI*/*abaR* QS system is widely present in *A. baumannii*. Tang J et al. collected 80 clinical isolates of *A. baumannii* from Jilin Province, China, from 2012 to 2017; 61 strains carried *abaI* and *abaR* genes, 24 of the 61 strains could produce AHL, and 76 of the 80 strains showed the ability to form biofilms [189]. In addition, they found that the *abaI* and *abaR* genes were positively correlated with bacterial drug resistance rates and surface-related motility. *AbaM* is a gene located between *abaR* and *abaI*; Lopez-Martin M et al. confirmed that Δ*abaM* mutant strains formed three times more biofilms than wild-type strains, and increased AHL production, indicating that *abaM* is a negative regulator of AHL production and biofilm formation [190]. QS also plays a role in the regulation of bacterial virulence factors. Sun X et al. evaluated the virulence of *A. baumannii* ATCC17978 with the *abaI*/*abaR* QS system. They found that Δ*abaI* and Δ*abaIR* mutant strains were completely nonvirulent, while the Δ*abaR* mutant remained virulent [191].

### 5.4. Other Mechanisms of Bacterial Resistance to Phages

Restriction-modification (R-M) systems that protect bacteria from bacteriophage infection can be classified into four categories: I−IV. The R-M system consists of genes encoding methyltransferase (MTase) and restriction enzyme (REase). Among them, MTase methylates host genome recognition sites, and REase recognizes unmethylated foreign genes and cleaves foreign DNA [192]. The R-M systems are the common defense system in bacteria, and Ambroa et al. identified genes associated with the R-M system in *A. baumannii* strains Ab2000 and Ab2010 [171]. When the phage gene injects into the host cell, the REase of the R-M system cleaves the phage gene by recognizing specific sites, while the host gene is not recognized by the Rease due to methylation by Mtase, thus resisting the phage infection [193].

Another bacteriophage-resistant strategy is abortive infection (Abi), in which bacteriophage-infected bacteria lyse their cells before phage maturation to protect other bacteria from infection [194].

The toxin-antitoxin (TA) system is widely present in *A. baumannii* [195,196,197] and is composed of toxin proteins and antitoxin proteins or antitoxin noncoding RNA [198]. Among them, toxins can regulate many important cellular processes in cells, which limits the growth of bacteria [199], and antitoxins can inhibit the activity of toxins. The antitoxin is unstable and easily degraded by protease. TA is believed to be associated with the resistance of *A. baumannii* [200] and is considered a promising drug target [201]. TA has also been shown to inhibit the biosynthesis of *A. baumannii* cell walls [202]. In addition, type IV TA has been found to mediate Abi, and thus play an anti-phage role [203]. Compared with *A. baumannii*, the anti-phage effect of TA has been studied more deeply in *Escherichia coli* [204,205]. Song S et al. found that the type I TA system Hok/Sok could inhibit phage T4 infection, and the possible mechanism of action is that phage T4 blocks the host transcription process, thereby blocking the production of Hok and Sok (antitoxin Sok is the single-stranded RNA). Because the antitoxin Sok is unstable and degraded first, the toxin Hok can play an anti-phage role [206].

At present, the CRISPR-Cas system is the most deeply studied anti-phage strategy in *A. baumannii.* However, R-M systems [207], Abi [208] and TA [206] systems, which have been extensively studied in other bacteria, have been studied relatively little in *A. baumannii*. However, it is believed that with increasing attention to phage therapy, research on the defense mechanism of bacteria against phages will continue to deepen, and the more specific action mechanism of *A. baumannii* against phage systems will be gradually clarified.

## 6. Phage Therapy for Drug-Resistant *A. baumannii*

With the abuse of antibiotics, MDR bacterial infections have emerged as a potential danger. Phages have been widely reported as a natural antibacterial agent, being used to deter the spread of MDR *A. baumannii* and to treat infections caused by *A. baumannii* [10,209]. Depending on the antibacterial mechanism, phage therapy can be classified as phage single therapy, phage cocktail therapy [59], the combination of phages with antibiotics, and application of phage products (lysozyme, depolymerase) (Figure 6).

### 6.1. Phage Single Therapy

Phages used as antimicrobial agents need to have the characteristics of a broad host range, short latent period, high burst size, and limited frequency of resistant bacteria [56,210]. Single-phage treatment is primarily used as a proof-of-concept for phage agents during design and testing. Although single-phage treatment has been successfully tested in animal models, data on clinical treatment are still lacking [211,212]. Phage vB_ABAP_PD-6a3, reported by Minle Wu et al., showed a lysis rate of approximately 32.4% against clinically resistant *A. baumannii*, which was approximately three times higher than the usual phage lysis rate [213]. However, the therapeutic effectiveness of single phages in the clinic may be reduced due to the existence of an arms race between resistant bacteria and phages [214]. Cocktail therapy is one of the most common methods in clinical treatment, as a single phage cannot always lyse various bacteria in the clinic.

### 6.2. Phage Cocktail Therapy

Cocktail therapy is a personalized treatment method that is often designed based on the specific conditions of patients. It targets a single bacterium or bacteria. Cocktail therapy is effective in preventing the emergence of resistant bacteria through a train of agents of different phages compared to simultaneous agents of phage mixtures [215]. Even if resistant bacteria appear during phage therapy, bacteria will be targeted by new phages that are added in the course of treatment. This strategy keeps the population density of bacteria at a low level for a long time and curbs the emergence of resistant bacteria [216].

### 6.3. Synergistic Effect of Bacteriophages and Antibiotics in A. baumannii

The main obstacle of bacteriophage therapy is the tolerance of bacterial evolution, which is faster than bacteriophage evolution. However, the evolution of bacteria to avoid phage lysis has resulted in certain sacrifices, including restoring sensitivity to antibiotics [217]. Based on this characteristic, the dual survival pressure exerted by the combination of bacteriophages and antibiotics can greatly slow the evolution of bacteria and provide sufficient clinical treatment time [168,169]. The φAB182 combined with antibiotics can eliminate the biofilm of MDR *A. baumannii* [218]. Therefore, the synergistic effect of bacteriophage and antibiotics on MDR bacteria is the most used and effective treatment strategy in clinical practice. A 42-year-old man’s left tibia was infected with MDR *K. pneumoniae* and MDR *A. baumannii* in a car accident. After treatment with the combination of antibiotics (colistin, Romepenem) and bacteriophage (bacteriophage AbKT21phi3, bacteriophage KpKT21phi1), the drug-resistant bacteria were completely eliminated, and the patient’s wound cured quickly [219]. The synergistic effect of bacteriophage and antibiotics has been proven to improve therapeutic efficiency in vivo and in vitro [51]. Pharmacological synergy occurs when the combined effect of multiple drugs is greater than the effect of a single drug, and it is typified by certain adjuvants that can block resistance mechanisms or improve drug pharmacokinetics. Antibiotics with appropriate concentrations can significantly improve the amount of phage lysis of bacteria. The reason is that lower concentrations of antibiotics inhibit the division of bacteria; therefore, the amount of biosynthesis of bacterial cells can be increased at this concentration [220,221]. In addition, antibiotics can promote phage lysis of bacteria, as morphological changes in bacteria cause changes in the peptidoglycan layer, which may lead to a higher sensitivity of bacteria to phage lyases (lysozymes, holins) [222]. A new approach was developed for the evaluation of *A. baumannii* phage-antibiotic synergic function [223]. The synergistic effect of these two effects (an increase in the number of phages and an improvement in the rate of bacterial lysis by phages) leads to superior results when phages and antibiotics work synergistically against drug-resistant bacteria.

### 6.4. Applications of Phage Enzymes

The bactericidal mechanism of phage enzymes is different from the mechanism of other agents. Compared to phages and antibiotics, phage enzymes are able to lyse a broader range of pathogenic bacteria without causing drug resistance. It is speculated that in the future, agents related to phage enzymes will be widely used for biological control, a strategy that will not only effectively contain the growth and spread of pathogenic bacteria, but also reduce the economic burden for patients [224]. The phage Petty, isolated by A.C. Hernandez-Morales et al., resulted in a clear halo around the center of the plaques, which suggests that phage Petty encodes a depolymerase that is capable of degrading the extracellular polysaccharides (EPS) of bacteria [152,225]. The enzyme was demonstrated to degrade the EPS of its host strain, AU0783, and reduce its viscosity. When the EPS of the bacteria is degraded, it will increase the accessibility of phage or other antimicrobial agents to the material inside the biofilm, and improve the efficiency of the bactericide. Hugo Oliveira et al. isolated a lytic phage, vB_AbaM_B9, of which ORF69 encodes a depolymerase (B9gp69) that degrades the EPS of *A. baumannii* ST-K30 and ST-k45, and does not cause drug resistance for the related bacteria [82]. Lysin packaged in hydrogel dressing is a safe and effective cure [226]. Furthermore, phage enzymes have some commercial value, as they can be protected by patents (which is different from phages). This advantage may facilitate the development and application of phage products in medicine, animal husbandry and food safety in the future [77,78,227].

## 7. Discussion

Phages are abundant in nature, and they can be obtained without complicated experimental manipulation and equipment, for experimental research or clinical practice. The technology of bacteriophage isolation is simple, and the experimental cost is low, which provides a basis for the industrial or clinical rapid formulation of personalized treatment. *In vivo*, bacteriophages can replicate and reproduce in their hosts, maintain high titers and reduce the adverse effects of multiple administrations.

Currently, phage therapy offers significant advantages in the context of a pandemic of MDR *A. baumannii*. However, there are also some limitations. The strong host specificity of phages makes it difficult to obtain pre-prepared phage cocktails, which need to be personalized for different patients [228], with a current high cost of the treatment [229]. First, in the absence of widespread theoretical knowledge of phage therapy, many patients disagree with the use of phages for treatment. This is one of the reasons why phage therapy is not approved by the FDA and EMA as a clinical drug treatment [230]. Furthermore, the uncertainties in the mode of administration, route, efficiency, timing, pharmacokinetics and immunomodulatory mechanisms of *in vivo* treatment of phages partially deter the clinical application of phage therapy [217]. In addition, the treatment of Gram-negative drug-resistant bacteria also requires consideration of endotoxin release, and the response of each person to endotoxin is very variable [231,232,233]. The survey data of Lynn El Haddad et al. showed that bacteriophages were effective (87%) and safe (67%), but only 35% of the researchers focused on the specific reasons why bacteriophages were resistant to host bacteria [234].

With the rapid spread and epidemic of MDR pathogens, it has become apparent that the development of antibiotics has failed to keep pace with the evolution of drug-resistant bacteria, and attention is being paid to phage therapy worldwide. Due to the diversity of phages, there are few studies of universal mechanisms based on phage and bacterial resistance, which has troubled in vivo studies of phage therapy. In the meantime, researchers are working individually and there is no effective organization to form a public phage bank to share resources and help build a system for the rapid preparation of personalized phages.

The maturation of next-generation sequencing technology and the development of genome editing technology provide the basis for phage modification and phage therapy. Based on second-generation sequencing technology, an abundant source of phages that exist in nature will be further exploited and utilized to provide a large number of candidates for clinical treatment. The development of gene editing technology enables further optimization of phages to be constructed, with engineered phages that meet the desired targets. Computer technology or artificial intelligence (AI) technology can also advance phage therapy based on phage genomic information, physiological properties, receptor binding and other information. In addition, the application of phage therapy in complex infections is promising. It has been shown to be effective in the treatment of patients with COVID-19 and secondary CRAB infection [235]. Alongside that, phage formulations are effective against challenging intracellular infections [236].

In this paper, we reviewed the current epidemiology and resistance mechanisms of drug-resistant *A. baumannii*, and described and discussed the current development of phages against drug-resistant *A. baumannii*. With the rapid spread of clinical MDR *A. baumannii*, phage therapy is a promising treatment for MDR *A. baumannii* infections. The combination of next-generation sequencing, gene editing and AI technologies makes it feasible to customize and optimize personalized phages, and there is no doubt that this will facilitate advances in phage therapy.

## Figures and Tables

**Figure 1 viruses-15-00673-f001:**
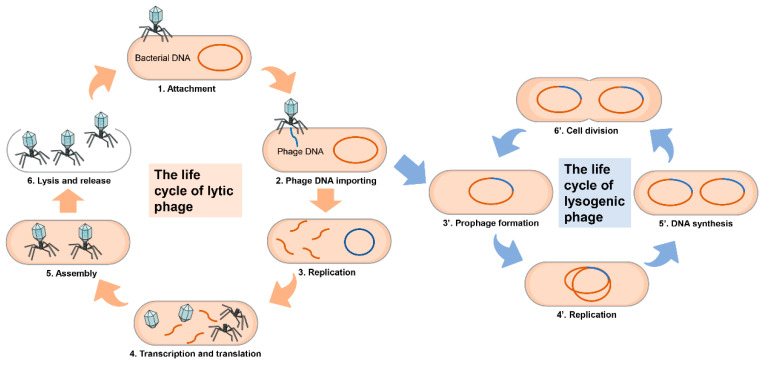
The life cycle of a phage can be cellular or lysogenic. The lytic cycle comprises: (1) the attachment of phage to the receptors on the cell membrane of host bacteria, (2) the import of its genome, (3) replication in the interior of the host cell, (4) subsequent transcription and translation, (5) assembly and (6) the release of phage progeny. Lysogenic phages enter the (3’) stage, are integrated into the bacterial chromosome and then start to replicate with the host cell.

**Figure 2 viruses-15-00673-f002:**
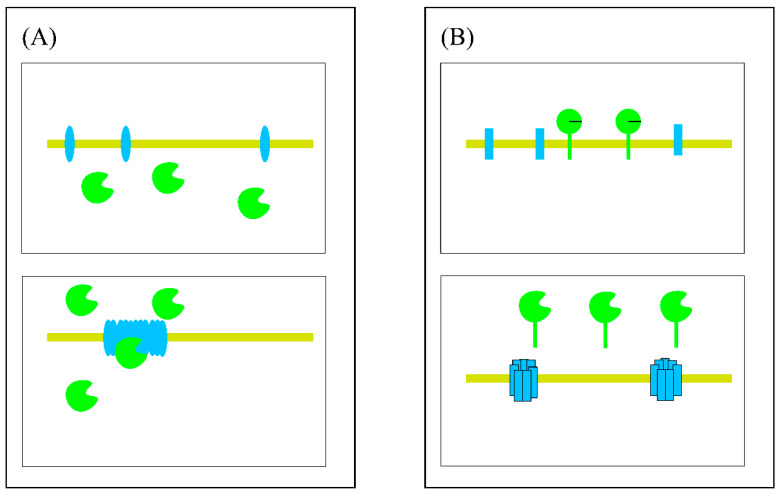
Two pathways to murein degradation. The inner membrane (IM) is shown by the yellow rectangle. (**A**) Classic holin–endolysin pathway: holin proteins (blue ovals) and endolysin (green ovals with open active sites). (**B**) SAR endolysin and pinholin pathway: pinholin (blue rectangles) and SAR-endolysin (green ovals with SAR domains). In the phage synthesis stage, holin or pinholin proteins are continuously synthesized and accumulate in the cytoplasm and IM in the form of a dimer. At the same time, endolysin protein is folded to form peptidoglycan-lytic enzymes and accumulates in the cytoplasmic matrix. SAR endolysin accumulates in the cytoplasm and IM without activity. When those proteins accumulate to a critical value, endolysin is released to lyse the peptidoglycan layer, and SAR endolysin refolds its conformation to obtain the active enzyme.

**Figure 3 viruses-15-00673-f003:**
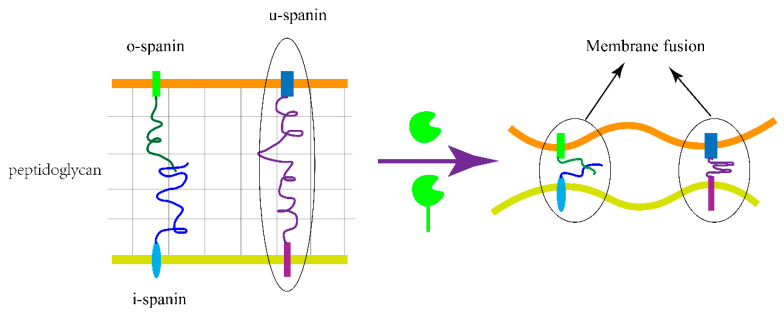
Two spanin systems disrupt the outer membrane by membrane fusion. The three spanins accumulate in the envelope and are trapped within the lacuna of the peptidoglycan layer. When endolysin or SAR endolysin removes the peptidoglycan layer, the spanins bring the two membranes together through conformational changes.

**Figure 4 viruses-15-00673-f004:**
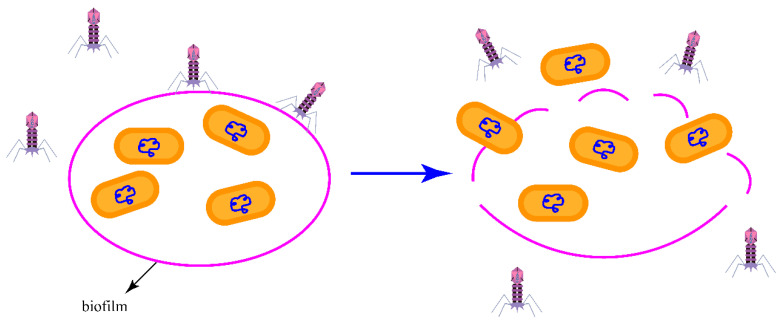
Phages act on biofilms, destroying the biofilm matrix, exposing receptors on the bacterial surface, and initiating the phage replication cycle. In addition, phage lysis of biofilms also increases the permeability to antibiotics, restoring the eradication effect of antibiotics on bacteria.

**Figure 5 viruses-15-00673-f005:**
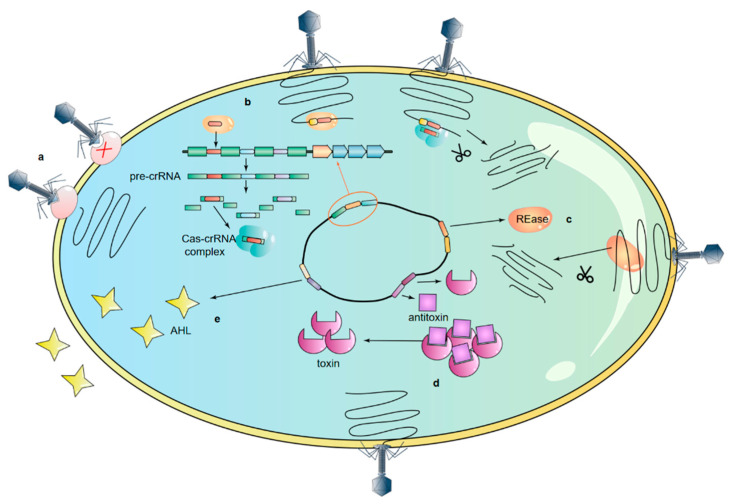
Bacteriophage resistance mechanism of *A. baumannii*. (a) Adsorption inhibition. Inhibition of phage adsorption by receptor deletion or structural change. (b) Inhibition of phage replication by CRISPR-Cas. (c) Restriction enzyme (REase) in the restricted-modification (R-M) system recognizes unmethylated phage genes and cleaves them. (d) The toxins in the toxin–antitoxin (TA) system can mediate abortive infection (Abi), and thus exert anti-phage effects. (e) *A. baumannii* resists phage infection through n-acyl-homoserine lactone (AHL) in quorum sensing (QS).

**Figure 6 viruses-15-00673-f006:**
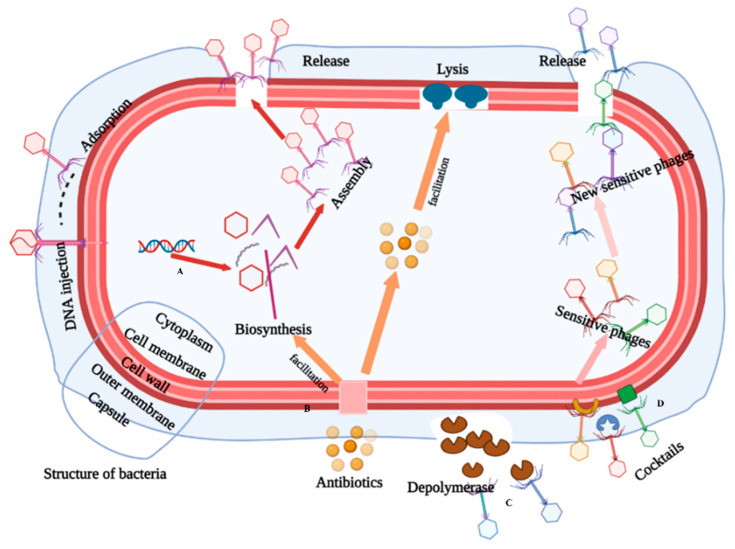
Phage therapy approaches. A Gram-negative bacterium that contains a cell capsule, outer membrane, and cell wall was mapped. A. Single-phage lytic bacteria; B. Combination of antibiotics and phages for use against bacteria (antibiotics facilitate the synthesis of phages). C. Phage enzyme against bacteria (depolymerase acts on extracellular polysaccharide, lysozyme acts on the cell wall). D. Cocktail therapy (multi-phage cooperative antibacterial).

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
