# Peer review of "Acinetobacter Baumannii Phages: Past, Present and Future"

_viruses, 2023, doi:10.3390/v15030673_

Round 1

Reviewer 1 Report

The provided review-manuscript is about “Acinetobacter baumannii phages: past, present and future”

I don’t think that it couldn’t be judged as a novel or significant enough to be published.

The abstract doesn’t reflect and summarize the background information and purpose stated in the manuscript title. Topics/paragraphs mostly don’t address to the titles given as well. Methods described are not adequately explained and synthesized with the statements made on.

Quite often statements/judging is not referred.

Data provided on other bacteriophages are not comprised, justified and synthesized/scientifically linked to Acinetobacter baumannii. Different data/texts are just scuttered in the manuscript body.

Numbering of headings and subheadings are mixed.

Conclusions and discussions are not deep and with scientifically arguments considering of Acinetobacter baumannii phages.

I wanted to give all relative comments but to addressed all of them is huge work and at the same time I don’t see usefulness of it. Just giving some:

Pages 12-14.

“such as β-lactams, quinolones, and polymyxins” – are no “mechanisms”.

“the genome of A. baumannii have been researching….” – needs to be rephrased.

Page 15. “Single phages are discovered, they may as an ingredient combine with

others to maximize the effect of treatment” – needs to be corrected.

Page 18. “we introduced the different drug resistance mechanisms” – statements is not correct.

Pages 25-38. Almost every sentence of “introduction” needs to be referred.

Pages 39-49. The same in the section “2”. And it doesn’t reflect the title of.

The sentence doesn’t sound scientifically correct: “The Antibiotics were once thought to be a highly effective class of chemicals that could treat bacterial infections.”

……………

Pages 135. “As the most numerous biological entities on Earth, there are approximately 1031phages” – sentence is repeated here.

Pages 149-229. needs to be referred.

New classification system should be used:

https://doi.org/10.3390/v13030506

Pages 231-235. Methods listed don’t correspond to the statement of “phage isolation and purification”.

Pages 230-293. Section “3.3” doesn’t reflect the title of.

References are not enough.

Page 399. “Here, we provide a more reasonable mode of action.”- it is confusing. Who provides/ based on what”?.

Page 411-418. Needs to be written in an argument-based manner and referred.

Page 424. Phages are temperate, condition/cycle is lysogenic -” Lysogenic bacteriophages”

Pages 440-441. By whom? - “gene related to drug resistance was detected and analyzed. Abouelfetouh et al. found that 8 of the 13 strains carrying the Oxa23 gene were located on”

Pagse 674-679. References need to be added.

Page 749. Not scientifically correct: “require only simple isolation”

Pages 751-752. Sentence is confusing: “and even multidrug-resistant bacteria can isolate

the required bacteriophages.”

……..

Author Response

Reviewer #1

I don’t think that it couldn’t be judged as a novel or significant enough to be published. The abstract doesn’t reflect and summarize the background information and purpose stated in the manuscript title. Topics/paragraphs mostly don’t address to the titles given as well. Methods described are not adequately explained and synthesized with the statements made on. Quite often statements/judging is not referred. Data provided on other bacteriophages are not comprised, justified and synthesized/scientifically linked to Acinetobacter baumannii. Different data/texts are just scuttered in the manuscript body. Numbering of headings and subheadings are mixed. Conclusions and discussions are not deep and with scientifically arguments considering of Acinetobacter baumannii phages. I wanted to give all relative comments but to addressed all of them is huge work and at the same time I don’t see usefulness of it. Just giving some:

Response: We really appreciate your time for reviewing and giving valuable suggestions. Here we give the responses respectively.

The Abstract section was revised and rephrased to “Acinetobacter baumannii (A. baumannii) is one of the most common clinical pathogens and a typical multi-drug resistant (MDR) bacterium. With the increase of drug-resistant A. baumannii infections, it is urgent to find some new treatment strategies, such as phage therapy. In this paper, we described the different drug-resistance of A. baumannii and some basic properties of A. baumannii phage, analyzed the interaction between phages and their hosts, and focused on A. baumannii phage therapies. Finally, we discussed the chance and challenge of phage therapy. This paper aims to provide a more comprehensive understanding of A. baumannii phages and theoretical support for the clinical application of A. baumannii phages.” (Line 11-20)

And we modified one of the keywords “antimicrobial resistance” to “Multi-drug resistance” (Line 20)

For the conclusion and discussion of section 7. We have added new evidence in the “7 Discussion” section.

We have rewritten the first sentence as “Phages are abundant in nature, and they can be obtained without complicated experimental manipulation and equipment for experimental research or clinical practice”. (Line 687-688)

We have added a new sentence after the sentence “The strong host specificity of phages makes it difficult to obtain pre-prepared phage cocktails, which need to be personalized for different patients [227]”, the new sentence is as follows “, with the current high cost of treatment [229].” (Line 696)

“In addition, the application of phage therapy in complex infections is promising. It has been shown to be effective in the treatment of patients with COVID-19 and secondary CRAB infection [235]. Besides, phage formulations are believed to be effective against challenging intracellular infections [236].” (Line 724-727)

Pages 12-14.

“such as β-lactams, quinolones, and polymyxins” – are no “mechanisms”.

“the genome of A. baumannii have been researching….” – needs to be rephrased

Response: Thanks for pointing out the mistakes

The phrases “such as β-lactams, quinolones, and polymyxins” and “the genome of A. baumannii have been researching….” have been rephrased in the abstract of revised manuscript.

Page 15. “Single phages are discovered, they may as an ingredient combine with

others to maximize the effect of treatment” – needs to be corrected.

Response: Thanks for pointing out our mistakes. We have rephrased the sentence in the abstract of revised manuscript.

Page 18. “we introduced the different drug resistance mechanisms” – statements is not correct.

Response: Thanks for pointing out the mistakes. We have rephrased the sentence in the abstract of revised manuscript.

Pages 25-38. Almost every sentence of “introduction” needs to be referred.

Response: Thanks for your valuable suggestion. We have added the relevant references, and rewritten the “introduction” as follows:

Acinetobacter baumannii (A. baumannii) is an essential gram-negative pathogenic bacterium widespread in nature [1]. A. baumannii can adhere to surfaces easily, as it has pods and pili [2]. Furthermore, since A. baumannii has strong invasive virulence factors, such as outer membrane proteins, lipopolysaccharides and phospholipases, the treatment of A. baumannii infection has been regarded as a great threat in clinical practice [3]. Antibiotics such as carbapenems, β-lactam antibiotics and polymyxins are commonly used clinically to treat A. baumannii infections [4-6]. However, the treatment of multi-drug resistant (MDR) A. baumannii is further aggravated by the abuse of antibiotics and the evolution of bacteria. Bacteriophages are bacterial viruses that specifically recognize, infect, and replicate within the host bacteria [7,8]. Phages have been considered as therapeutic agents since the early 1920s as a result of their unique antibacterial ability. In addition, phages have the advantages of strong antibacterial ability, high quantity (1030- 1032 in the earth), and low toxic side effects to humans, and are considered as one of the most promising drugs to replace traditional antibiotics [9,10].” (Line 23-36)

Pages 39-49. The same in the section “2”. And it doesn’t reflect the title of.

The sentence doesn’t sound scientifically correct: “The Antibiotics were once thought to be a highly effective class of chemicals that could treat bacterial infections.”

Response: Thanks for the valuable suggestion about the title. We have revised the original title to “Antibiotic resistance in Acinetobacter baumannii”. (Line 37)

The previous sentence “The antibiotics were once thought to be a highly effective class of chemicals that could treat bacterial infections. However, with the abuse of antibiotics, drug-resistant bacteria appear constantly, leading to a huge challenge for human health and safety. However,...” has been replaced as “Drug-resistant bacteria continue to emerge, posing a huge challenge to human health and safety.” (Line 38-39)

In response to the lack of references, we have added new references in this section, “Bacteria can develop antimicrobial resistance (AMR) to a variety of antibiotics, such as β-lactams [12], quinolones [13] and polymyxin [14], through intrinsic resistance mechanisms, such as increased efflux pumps [15], decreased outer membrane proteins (OMPs) [16] and acquired resistance mechanisms [14]. In addition, phages can also mediate antibiotic resistance in bacteria through transduction”. (Line 42-46)

Pages 135. “As the most numerous biological entities on Earth, there are approximately 1031 phages” – sentence is repeated here.

Response: Thanks for pointing out our mistakes.

We have deleted “There are approximately 1031 phages”.

Pages 149-229. needs to be referred. New classification system should be used:

Response: Thanks for pointing out the mistakes and the lack of references. We have inserted the new references and rewritten with new references mainly according to the newest standard of ICTV (https://doi.org/10.1007/s00705-022-05694-2). And the new paragraph is as follows:

“The viral nature of phages was controversial until the early 1940s, when they were observed by electron microscopy, confirming their particulate nature and enabling them to be classified based on morphology [45], the main basis for the current classification of phages. Classic electron microscopy images are formed by atoms of heavy metals such as uranium, which evaporate in a vacuum and allow for the sample to be struck at an angle. The introduction of negative stains (heavy metal salts that dry in thin layers, do not form crystals and can embed small particles such as phages) in electron microscopy has resulted in more detailed images than earlier methods and revealed the complexity and diversity of phage morphology [46]. Assigning phages to different taxonomies is a fundamental step in phage research. As more and more new phages are discovered, ICTV's classification criteria are constantly changing. The most recent standard is the August 2022 phage classification system, which removes several major families in the order Siphoviridae, Podoviridae and Myoviridae. But the classical description of its morphology as belonging such as "podovirus", "myovirus", or "siphovirus" remains. The order “Caudovirales” was also deleted and replaced by the class Caudoviricetes. And establishing a binomial system of nomenclature for species [47].” (Line 151-165)

Pages 231-235. Methods listed don’t correspond to the statement of “phage isolation and purification”.

Response: Thanks for pointing out the mistakes. We have deleted the paragraph.

Pages 230-293. Section “3.3” doesn’t reflect the title of. References are not enough.

Response: Thanks for pointing out the problem. We changed a lot in the previous section 3.3. and added references. And the title of 3.3 was modified to “3.3 Life Cycle and biological properties of phages”. (Line 187). We have rephrased and added references to the section “3.3” as follows:

“Phages can be divided into temperate phages and lytic phages, which have different life cycles (Figure 1). Infection begins with the adsorption of phage, which interacts with specific receptors on the surface of host cells, and then the phage injects its genome materials into the cytoplasm of the host, next, transcription and replication occur. Subsequently, once the synthesis and assembly of viral proteins are completed, the phage genome materials are packaged into the capsid. Phage-encoded depolymerase hydrolyses the peptidoglycan layer of bacterial cells, leading to cell lysis and the release of mature virions. The phage progenies are released into the environment and then attach to the next phage-sensitive receptor [60]. Temperate phages can enter either lytic or lysogenic cycles [61], for example, λ phage, after the injection of DNA into the cytoplasm, the temperate phage could choose to initiate a lytic cycle, which would be consistent with the virulent phage. Either it integrates its DNA into the host bacterial chromosome, or at the same time, the viral genome is named prophage. This phage expresses a specific phage repressor that represses transcription factors, including lytic cycle genes, and thus the phage enters a dormant state [62]. The integrated phage dsDNA replicates with the bacterial chromosome during cell division and thus is passed down through generations in bacteria. When the phage is excised from the host chromosome, it exits the lysogenic cycle [63].” (Line 189-205)

Page 399. “Here, we provide a more reasonable mode of action.”- it is confusing. Who provides/ based on what”?

Response: Thank you for your kind reminder. We have changed the sentence to “Young, R provides a reasonable mode of action [103].” (Line 322-332)

Page 411-418. Needs to be written in an argument-based manner and referred.

Response: Thank you for your constructive suggestions. We have added references to support the statements.

“Generally, the phage cleavage process of gram-negative bacteria needs to break through three barriers: the intima, peptidoglycan layer and outer membrane. The cleavage of the intima and peptidoglycan layer is accomplished by two groups of matching proteins, namely, classical holin-endolysin and SAR-endolysin/pinholin [103]. On this basis, the cleavage of the outer membrane at the spanin protein is an essential step [104]. At present, the characterized spanin system is divided into a u-spanin one-component system and an i-spanin and o-spanin two-component system. In theory, there are four different combinations of bacteriophages in the cleavage of gram-negative bacteria.” (Line 340-347)

Page 424. Phages are temperate, condition/cycle is lysogenic -” Lysogenic bacteriophages”

Response: Thank you for your kind reminder. We have added this sentence to make the expression more clearly.

“The life cycle of temperate phage is lysogeny.” (Line353-354)

Pages 440-441. By whom? - “gene related to drug resistance was detected and analyzed. Abouelfetouh et al. found that 8 of the 13 strains carrying the Oxa23 gene were located on”

Response: Thank you for your kind reminder. We have changed this sentence to make the expression more clearly.

“Based on the genomic analysis of the current pandemic carbapenem-resistant A. baumannii, the Oxa23 gene related to drug resistance was detected and analyzed by Abouelfetouh et al. The results have shown that 8 of the 13 strains carrying the Oxa23 gene were located on the genome of the phiOXA prophage [110].” (Line 369-372)

Pages 674-679. References need to be added.

Response: Thanks for your valuable suggestion. We have added the relevant references, and the details are as follows:

“With the abuse of antibiotics, MDR bacterial infections have emerged as a potential danger. Phages have been widely reported as a natural antibacterial agent, being used to deter the spread of MDR A. baumannii and to treat infections caused by A. baumannii [10,208]. Depending on the antibacterial mechanism, phage therapy can be classified as phage single therapy, phage cocktail therapy [59], the combination of phages with antibiotics, and application of phage products (lysozyme, depolymerase) (Figure 6).” (Line 612-617)

Page 749. Not scientifically correct: “require only simple isolation”

Response: Thanks for pointing out our mistakes. We have modified “require only simple isolation” to “Phages are naturally abundant in nature, and they can be obtained without complicated experimental manipulation and equipment for experimental research or clinical practice.” (Line 687-688)

Pages 751-752. Sentence is confusing: “and even multidrug-resistant bacteria can isolate the required bacteriophages.”

Response: Thanks for your reminder and consideration. We have deleted this sentence.

Reviewer 2 Report

Hello Authors,

I am very happy to see the review manuscript. I have some comment. Please consider that.

Line 49 …no reference ??

Line 50 …the paragraph including information is not sufficient

Line 88-89 …I don’t understand

Line 101 ..There is no information about polymyxin B

Line 131….Heading number 3 but previous one 3.2 ??? What is correct one ?

Line 149 .. Structural classification and genomics? Proper reference needed and insufficient information.

Phage receptor binding site did not inform with reference and their mechanism lacking .

Phage resistance mechanism needed more information.

Future direcetion needed with more evidence.

I did not find any significant value of this review. Please rewrite with significant evidence in terms of phage therapy.

Thank you.

Author Response

Reviewer #2 (Comments and Suggestions for Authors)

Hello Authors, I am very happy to see the review manuscript. I have some comment. Please consider that.

Line 49 …no reference ??

Response: Thank you very much for your suggestion, references have been added to this section. The result of the modification is as follows, “Bacteria can develop antimicrobial resistance (AMR) to a variety of antibiotics, such as β-lactams [12], quinolones [13] and polymyxin [14], through intrinsic resistance mechanisms, such as increased efflux pumps [15], decreased outer membrane proteins (OMPs) [16] and acquired resistance mechanisms [14]. In addition, phages can also mediate antibiotic resistance in bacteria through trans-duction”. (Line 42-46)

Line 50 …the paragraph including information is not sufficient

Response: Thank you for your valuable comments. I think you are referring to the first paragraph of “2.1 β-lactam class”. We have modified this paragraph.

“β-lactam antibiotics (BLAs) are widely used in the clinical treatment of A. baumannii infection because of their ability to act on penicillin-binding proteins (PBPs) and inhibit the synthesis of cell walls [17]. However, the hydrolysis of BLAs by β-lactamase [18] and the structural changes of PBPs seriously affect the clinical efficacy of BLAs [19]. Carbapenem antibiotics include imipenem (IPM) and meropenem, of which IPM is the first highly effective broad-spectrum carbapenem [20]. Carbapenems are atypical BLAs and are considered to be one of the first choices for the treatment of A. baumannii infection [21]. However, since the first carbapenem-resistant OXA-type β-lactamase (blaOXA-23-like enzyme) was discovered in A. baumannii strains, many β-lactamases that can hydrolyze carbapenems have continuously been discovered, such as blaOXA-51-like enzyme and blaOXA-58-like enzyme [22]. In addition, the reduction of OMPs can also lead to an increase in the resistance of A. baumannii to carbapenems [23]. With the increasing resistance rate of carbapenem-resistant A. baumannii (CRAB), how to treat CRAB has become a difficult problem worldwide. Previous studies have demonstrated that extensively drug-resistant (XDR) or pandrug-resistant (PDR) CRAB can lead to high morbidity and mortality, and the carbapenem antibiotic resistance rate has reached 90% in some regions [24]. The World Health Organization has identified CRAB as a prime pathogen for urgent drug development”. (Line 48-65)

Line 88-89 …I don’t understand

Response: This is an unrelated text in the template, we are very sorry it appeared in the article and we have deleted it.

Line 101. .There is no information about polymyxin B

Response: Thank you very much for your comments. To make the content of the article more relevant to polymyxin B (PMB), we have modified the sentence from “Polymyxin B (PMB) can increase the permeability of the bacterial outer membrane (OM) and destroy its structure, and is considered the last line of defence against drug-resistant A. baumannii” to “Polymyxin B (PMB) is considered to be the last-line of defense against drug-resistant A. baumannii [28]. It changes the bacterial outer membrane (OM) charge by interacting with lipid A, causing an increase in the permeability of the OM and a disruption of the OM structure, thereby achieving the purpose of sterilization [29].” (Line 93-96)

And we have added the new paragraph “Zhao et al. demonstrated that the level of resistance evolution of A. baumannii to PMB is related to the concentration of PMB, and that higher concentrations of PMB are more favorable for the evolution of bacterial resistance [31].” (Line 107-110)

Line 131….Heading number 3 but previous one 3.2 ??? What is correct one ?

Response: We are sorry for the error, the title number has been changed. The correct title is as follows: “2.2. Quinolone class; 2.3 Polymyxin class; 2.4 Phage-mediated antibiotic resistance”.

Line 149 .. Structural classification and genomics? Proper reference needed and insufficient information.

Response: Thanks for your suggestion, in this section we have added references and revised. The new title is “3.2 Structure and genomics”.

We have rewritten new paragraphs with new references mainly according to the newest standard of ICTV (https://doi.org/10.1007/s00705-022-05694-2). The rewritten contents are as follows.

“3.2 Structure and genomics

Normally, the head of the phage is prismatic and the single genetic material (DNA/RNA) is contained with enveloped by protein. Attached below the head is the neck or collar region of the elongated sheath (sometimes called the tail). The DNA/RNA is injected into the host cell through its internal hollow structure and is surrounded by protective sheath proteins [38]. The base of the sheath is a baseplate to which are attached tail fibers which is a key structure for attachment and infection of host cells, its function is primarily to recognize the surface receptor of the host cell and complete the infection process by binding to the host cell receptor. The tail fiber protein is typically composed of multiple subunits and has high variability, which allows the formation of diverse tail fiber structures by different combinations of subunits [39], enabling the infection of different host cells, thereby tail fiber proteins can be used in the detection of A. baumannii [40-42]. The special cases include filamentous phage [43] and barely non-tailed phage [44]. At present, there are no definite reports that the host of filamentous phage is A. baumannii.

The viral nature of phages was controversial until the early 1940s, when they were observed by electron microscopy, confirming their particulate nature and enabling them to be classified based on morphology [45]. Classic electron microscopy images are formed by atoms of heavy metals such as uranium, which evaporate in a vacuum and allow for the sample to be struck at an angle. The introduction of negative stains (heavy metal salts that dry in thin layers, do not form crystals and can embed small particles such as phages) in electron microscopy has resulted in more detailed images than earlier methods and revealed the complexity and diversity of phage morphology [46]. Assigning phages to different taxonomies is a fundamental step in phage research. As more and more new phages are discovered, ICTV's classification criteria are constantly changing. The most recent standard is the August 2022 phage classification system, which removes several major families in the order Siphoviridae, Podoviridae and Myoviridae. But the classical description of its morphology as belonging such as "podovirus", "myovirus", or "siphovirus" remains. The order “Caudovirales” was also deleted and replaced by the class Caudoviricetes. And establishing a binomial system of nomenclature for species [47].

Genome sequencing revealed the abundance of the prophage. Comparative genomics showed the co-evolutionary relationship between phages and their host bacteria as an essential tool to reveal phage diversity [48], and provide the strategy of therapy [49]. The framework of the A. baumannii phage genome is mainly composed of genes from other phages, with a relatively small proportion of genes from the host A. baumannii [50]. Morphogenesis of the tail occurs frequently, the mutant Ab105-2phiΔCI404ad, genomic rearrangement increases the host range of the phage nearly 3-fold [51]. There is no genome similarity of vB_AbaS_TCUP2199 to other known phages [52]. It has been documented that most of the genes of bacteriophages are unknown [53]. Their genome size range is wide, and the structure of their genome is linear or circular [54]. The most common type of genome is dsDNA among known phages. The genome size of phage KARL-1 was determined to be 166,560 bp, A total of 253 ORFs were identified, involved in the replication, maturation and release of phage progeny, some encoded structural proteins, most ORFs encoded hypothetical phage-like proteins, and the rest are hypothetical proteins [55]. The genome of A. baumannii phage Abp9 contains 80 ORFs but lacks any known virulence genes or lysogen-forming genes [56]. The two phage strains, WCHABP1 and WCHABP12S, contain seven structural proteins. In addition, both encode a gene for a protein containing lysozyme that is also possessed by other phages of the genus Ap22virus [57].” (Line 138-184)

Phage receptor binding site did not inform with reference and their mechanism lacking .

Response: Thanks to your constructive suggestion, in this section we have added references and new sentences, “the capsule of A. baumannii is the most common receptor for phage [79,165,171,172]” (Line 484-485)

and “Similarly, single-base deletions in the gtr6 gene which encodes glycosyltransferase in A. baumannii AB5001 resulted in a change in the structure of the K3-type capsular polysaccharide (CPS), thereby inhibiting the ability of the phage to infect the bacteria [171].” (Line 493-496)

Phage resistance mechanism needed more information.

Response: Thank you very much for your suggestion on section “5”, we have added new references in this section. In addition, in the first paragraph of section “5.2”, we have changed the original sentence. (Line 509-517)

In this section of 5.4, we have added new sentences to clarify phage resistance mechanism “The R-M systems are the common defense system in bacteria, and Ambroa et al. identified genes associated with the R-M system in A. baumannii strains Ab2000 and 2010 [170]. When the phage gene injects into the host cell, the REase of the R-M system cleaves the phage gene by recognizing specific sites, while the host gene is not recognized by the REase due to methylation by MTase, thus resisting the phage infection [192].” (Line 581-586)

Future direction needed with more evidence.

Response: Thanks for your constructive suggestion.

We have added new evidence to provide some future direction in “7 Discussion section”.

We rewrote the first sentence “Phages are naturally abundant in nature, and they can be obtained without complicated experimental manipulation and equipment for experimental research or clinical practice”. (Line 687-688)

Line 745-748 “In addition, the application of phage therapy in complex infections is promising. It has been shown to be effective in the treatment of patients with COVID-19 and secondary CRAB infection [235]. Besides, phage formulations are effective against challenging intracellular infections [236].” (Line 724-727)

I did not find any significant value of this review. Please rewrite with significant evidence in terms of phage therapy.

Response: Thanks for your valuable suggestion. We have rewritten with significant evidence in terms of phage therapy. We added sentences with references:

“With the abuse of antibiotics, MDR bacterial infections have emerged as a potential danger. Phages have been widely reported as a natural antibacterial agent, being used to deter the spread of MDR A. baumannii and to treat infections caused by A. baumannii [10,208].” (Line 612-617)

“The φAB182 combined with antibiotics can eliminate the biofilm of MDR A. baumannii [218].” (Line 645-646)

“A new approach was developed for the evaluation of A.baumannii phage-antibiotics synergic function [223].” (Line 661-662)

“Lysin packaged in hydrogel dressing is a safe and effective cure [226].” (Line 681-682)

Round 2

Reviewer 1 Report

Pages 14 -18.

The title is: “Acinetobacter baumannii phages: past, present and future”

And abstract says:

In this paper, we described the different drug-resistances of A. baumannii and”

Why some? - some basic properties of A. baumannii phage, analyzed the interaction between phages and their hosts, and focused on A. baumannii phage therapies.

Then it is repeating again - Finally, we discussed the chance and challenge of phage therapy. This paper aims to provide a more comprehensive understanding of A. baumannii phages and theoretical support for the clinical application of A. baumannii phages.

Pages 22 -36. Introduction doesn’t reply the manuscript title and content.

Pages 32 -33.

 “Phages have been considered as therapeutic agents since the early 1920s as a result of their unique antibacterial ability.

Then again – “In addition, phages have the advantages of strong antibacterial ability,

Pages: 37-125

here, the details given about antibiotic resistance are not further discussed in relation with phage therapy strategies/supports development as authors state in the abstract and doesn’t correspond to manuscript title as well.

Pages: 130-136

This section doesn’t say anything that is specific to A. baumannii phages and doesn’t give message in regards of manuscript title.

Pages 136-184.

The half of the section doesn’t particularly say about A. baumannii phages.

The rest presents kind of list of A. baumannii “prophages” whiteout making some reasonable interpretation related to phage therapy/title of manuscript.

Pages 186-187.

“Phage growth is determined” by many parameters – it is not sentenced correctly

“namely, the phage bacterial adsorption rate constant, burst size, latent period, bacterial growth rate, phage and bacterial elimination rates, and the effect of controlled release of phage” – needs to be revised.

Pages 189-205.

This section doesn’t say anything that is specific to A. baumannii phages and doesn’t give message in regards of manuscript title.

Pages 206-215.

Here, is given some information about A. baumannii phages parameters without interpretation, discussions.

………………

Data provided on Acinetobacter baumannii and in general on bacteriophages are not comprised, justified and synthesized. Different data/texts are just scuttered in the manuscript body.

Reviewer 2 Report

Go ahead